# Determination of Glyphosate in White and Brown Rice with HPLC-ICP-MS/MS

**DOI:** 10.3390/molecules27228049

**Published:** 2022-11-19

**Authors:** Maria Chiara Fontanella, Lucrezia Lamastra, Gian Maria Beone

**Affiliations:** Department for Sustainable Food Process, Università Cattolica del Sacro Cuore, Via Emilia Parmense 84, 29122 Piacenza, Italy

**Keywords:** glyphosate, HPLC-ICP-MS/MS, white and brown rice, LOD, LOQ, matrix effect

## Abstract

Background: In 2017, the European Commission renewed the approval of glyphosate (GLY) but only for five years. GLY remains one of the most controversial and studied molecules. Method: A simplified method was tested for the determination of GLY in white rice (WR) and brown rice (BR), after extraction only with a methanol solution, by liquid chromatography coupled with inductively coupled mass triple quadrupole (HPLC-ICP-MS/MS) with a PRP-X100 anionic column. After performing a test on groundwater, the quantification of GLY in WR and BR was validated in terms of the LOD, LOQ, accuracy, precision, linearity, and the matrix effect. Results: The LOD was 0.0027 mg kg^−1^ for WR and 0.0136 mg kg^−1^ for BR. The LOQ was 0.0092 mg kg^−1^ for WR and 0.0456 mg kg^−1^ for BR. The mean recoveries were within 76–105% at three fortification levels. The relative standard deviation for the analysis (five replicates for three spike levels) was < 11% for both matrices. A linear response was confirmed in all cases in the entire concentration range (R^2^_WR_ = 1.000 and R^2^_BR_ = 0.9818). Conclusion: The proposed method could be considered useful for the determination of GLY in different types of rice and designed and adapted for other cereals. The matrix effect, quantified in BR matrix extraction, could be avoided by using a matrix-matched calibration line.

## 1. Introduction

Glyphosate (GLY) was approved in 2002 for the first time at the EU level for ten years under a directive replaced in 2011 by Regulation 1107/2009 [1], but it had been on the market in most countries under national legislation since the 1970s [2].

Glyphosate is the ISO common name for N-phosphono methyl glycine (the IUPAC name). It is a small molecule with three functional groups that make it highly polar, stable, and soluble in water. Moreover, it is easily absorbed by the main components of the soil, such as clays, iron oxides, and humic substances. Combined with GLY, aminomethylphosphonic acid (AMPA) is considered an indicator of the application of the GLY molecule, because it is a product of its degradation and has similar chemical characteristics to its precursor. This molecule is an active substance in plant protection products, and their products are authorized for a wide variety of crop and non-crop uses, for example, on railways, hard surfaces, etc. Its application occurs either pre-sowing, preharvest, or postharvest, due to its unparalleled action on annual and perennial weeds [3].

The debate on using GLY in Europe is still very active in terms of human health. In 2015, the International Agency for Research on Cancer (IARC) and the European Food Safety Agency (EFSA) put forward two conflicting views on the carcinogenic hazard of GLY to humans. In 2016, it blocked a 15-year renewal proposal for the use of this molecule. In 2017, the European Chemical Agency (ECHA) classified GLY as non-carcinogenic. In the same year, on 12 December 2017, the European Commission renewed the approval of GLY but only for five years. It will be up for assessment again in 2022 [2,4]. Before considering the decisions concerning the danger of this molecule, environments such as streams and groundwater are being impacted by the application of GLY, and European legislation established maximum residue limits, 0.1 μg L^−1^, in water for human consumption with the Directive 1998/83/EC [5]. In the USA, the maximum contaminant level of GLY is 700 μg L^−1^ [6].

Water is an essential element in the grain food chain (irrigation, food processing, food intake, etc.) and is a source of contamination with herbicides. Myers et al. (2016) [7] demonstrated that glyphosate-based herbicides had contaminated drinking water and agricultural systems. In 2013, the European Commission uploaded maximum residue levels (MRLs) for GLY in several crops according to European regulation EU No. 293/2013 [8]. The new European MRLs for rice are 0.1 mg kg^−1^, as the lower limit of analytical determination. On some cereals, such as rice, where minimal processing is desired, and consumption of the outer coat is recommended for associated health benefits [9], further studies should also be conducted for GLY in grains at different cleaning stages.

Several methods are available for the determination of this molecule and its metabolites in different kinds of matrices, such as apples [10], cereals [11], soybean [12,13], and surface and wastewater [14,15]. GLY and its metabolite have been determined by different analytical techniques, including GC [10,13,16,17], LC [13,15,16,18,19,20], hydrophilic interaction chromatography [21], capillary electrophoresis [22,23], Flow-Injection-MS/MS [24,25], GC-MS/MS, and LC-MS/MS [10,13,16,26,27,28,29,30,31,32], but the chemical characteristics, such as solubility and polarity, have been an obstacle with these techniques. Derivatization and purification of the molecules are methods for detection and chromatographic separation of the compounds in the matrices, but these actions are time-consuming and costly. The numerous sample preparation steps have increased interest in developing new methods to reduce the number of actions, from sample collection to analysis, and to reduce the cost. 

Different authors have described GLY determination combined with AMPA and/or other phosphorus herbicides with HPLC–single quadrupole ICP-MS [14,33,34,35,36]. From 2019, in the literature, the number of articles showing applications of triple quadrupole with inductively coupled plasma and HPLC are few, and they show applications on aqueous samples. Lajin and Goessler (2019) [37] developed an HPLC-ICP-MS/MS speciation analysis method to determine this pesticide in tap, ground, and river water. Subsequently, Tiago et al. (2020) [38] also observed the exudation process of GLY molecules in water from hydroponic systems. In the same year, Pimenta et al. [39] compared two detection systems coupled to HPLC in quantifying GLY and AMPA in natural water samples from artesian wells, dams, water springs, and cisterns.

Applying ICP-MS/MS with HPLC improves the selectivity and the detection limits compared to the single quadrupole ICP-MS, due to its elimination of polyatomic interferences on m/z 31. The technology is based on two quadrupoles and a reaction cell; the first quadruple could be used as a filter to select a specific m/z, and after the reaction with a specific gas, the second quadruple could be applied as a selector. Even if the detection of phosphorous is highly compromised by its high ionization potential, the reaction cell can chemically resolve ^31^P^+^ from polyatomic ions by its oxidation to ^31^P^16^O^+^ after preferential reaction with oxygen gas supplied to the cell by shifting analytes to new product ion mass [40].

Our attention is focused on the development and validation of ionic determination based on liquid chromatography with the inductively coupled plasma tandem mass of molecules of GLY in different types of rice (white and brown rice), based on rapidity and simplicity.

## 2. Results and Discussion

In order to develop the HPLC-ICP-MS/MS for the detection of GLY and AMPA, the first experiments were realized to select the optimal reaction gas flux, instrument parameters, appropriate mobile phase concentration, and column characteristics in the analysis of ultrapure water with standard addition. In order to study how it is possible to quantify GLY in more complex solutions and an eventual matrix effect, the method was applied in groundwater spiked with GLY and AMPA (Figure 1).

The LOQ, determined as the lowest fortification level for GLY (1.8 μg L^−1^, with a mean recovery equal to 99% and an RSD equal to 2%), was 0.29 μg GLY L^−1^ in groundwater. Tiago et al. (2020) [38] published an LOQ value of 1.09 μg GLY L^−1^ in water samples. Lajin and Goessler (2019) [37] and Pimenta et al. (2020) [39] found 1.05 μg GLY L^−1^ and 8.2 μg GLY L^−1^, as detection limits in their respective method. These cited authors used the same type of analytical instrument, HPLC-ICP-MS/MS. One of the most important results in our early stage of method development was that our limit of quantification in groundwater was lower than the maximum residue limit, 0.1 μg L^−1^, established in the Directive 1998/83/EC [5] in water for human consumption.

Before applying this analytical method to rice, the matrix effect was studied to compare and understand when the quantification of our molecules in groundwater differed from the quantification of the same content in the mobile phase with this analytical technique. We noticed a matrix effect of 92% by comparing the calibration curve realized with the standard addition in groundwater to the same calibration curve constituted in the mobile phase, proving that there was a matrix suppression effect. The study of the matrix effect is a crucial point when starting to study more complex matrices. The same configuration was applied to discriminate compounds in a more complex matrix, such as a cereal. The composition of the mobile phase remained the same. Only the concentration of malonic acid was diluted to ensure that the GLY peak was as far away from the phosphate peak as possible. Of course, the tests on the groundwater were useful to find satisfactory parameters for both instrumental and chromatographic areas to start with an adequate background. An important parameter, which allowed improvement of the quantification, was the application of an anionic exchange chromatographic column with Poly (styrene-divinylbenzene) with a tri-methyl ammonium exchanger-like support material, the option to use up to 60° at a pH between 1 and 7.9, and a hardware inner diameter equal to 2.1 mm. The same type of chromatographic column with a greater inner diameter (4.6 mm) did not adequately quantify 0.05 mg GLY kg^−1^ in WR, with a low recovery (54% ± 11%).

The extraction solution was based on an aqueous composition. We changed the acid component, introducing different percentages of formic acid, because the retention time (RT) of GLY was not stable during the time of analysis with the application of only ultrapure water. Unfortunately, the formic acid strongly altered the repeatability conditions of the analysis from one sample to another, especially the background line of the chromatogram. The choice of methanol was suggested in numerous articles that aimed to analyze GLY in food matrices. The European Reference Laboratory (EURL, EU Reference Laboratory) developed a generic method named the Quick Polar Pesticides Methods (QuPPe) [41] based on the extraction of polar pesticides from a sample with acidified methanol and liquid chromatography coupled to tandem mass spectrometry (LC-MS/MS). This was used to determine polar pesticides in different foods [42]. Some authors applied an aqueous extraction with methanol, such as Nagatomi et al. (2013) [27] on beer, barley tea (liquid), malt, and corn, or with acidified methanol, such as Herrera Lopez et al. (2019) [32] on grapes, orange, lettuce, oat, and soya beans. Those publications performed GLY determination by LC-MS/MS (Q-TRAP hybrid triple quadrupole) and LC-ESI-QTRAP-MS, respectively. 

The validation parameters were the accuracy, precision, LOQ, LOD, and the matrix effect. An external calibration solution was employed to quantify the GLY.

The trueness in terms of the recovery percentage was calculated for the GLY at three fortification levels, 0.01, 0.03, and 0.05 mg kg^−1^ from WR and 0.05, 0.14, and 0.27 mg kg^−1^ from BR. The average recoveries ranged from 76 to 105%, with RSDr <15% for all spiking levels studied and for white and brown rice. The trueness and precision were within the specifications, because the obtained percentages were between 70 and 120%, and the RSD values were less than 20%, as recommended by the main legislations [43]. Table 1 shows the recoveries, standard deviation, and relative standard deviation for both types of rice. Figure 2 and Figure 3 show the chromatogram of the enriched samples at the spiked level of 0.01 mg GLY kg^−1^ in the WR (Caravaggio variety) and 0.1 mg GLY kg^−1^ in the BR (Barone variety).

### 2.1. Linearity and Matrix Effect

The linearity was evaluated for GLY in the extraction solution (aqueous solution with 30% of methanol) and in the extraction solution after application of the extraction solution on different rice matrices. The range concentration of the calibration curves applied to study the linearity in the WR solution, with standard additions, ranged from 0.003 to 0.055 mg GLY kg^−1^ (Figure 4). 

The range concentration of the calibration curves applied to study the linearity in the BR solution, with standard additions, ranged from 0.11 to 0.55 mg GLY kg^−1^ (Figure 5). 

Good correlation coefficients were obtained for both types of calibration curves. The calibration curves demonstrated that the detector response was linear in the concentration range for both types of rice.

Moreover, in Figure 4 and Figure 5, it was possible to observe and calculate the matrix effect in the WR and BR extraction solution on the GLY determination. The matrix effect depends on the instrument used for the analysis, the type of matrix, the properties of the molecules, and the analyte concentration. Consequently, the matrix effect can vary for each analyte and matrix, which was exactly what we observed. The ratios between the slopes obtained from the analytical curve in the matrix and only in the extraction solution were expressed in percentages. The corresponding values were 96% and 80% for GLY quantification in WR and BR, respectively. The first value revealed a nonsignificant matrix effect in determining this molecule in the white rice extraction solution by HPLC-ICP-MS/MS. A significant matrix effect was recorded when the BR became the extraction object, showing a matrix suppression on its analyte. This value was very near to the MEs calculated in rice by Botero-Coy et al. (2013) [26] and Santilio et al. (2019) [31]. In particular, in this case, using a matrix-matched calibration line was necessary to obtain more accurate results and avoid the underestimation of the concentration of GLY in rice. Table 2 shows that the GLY in cereals was quantified with LC tandem MS in most articles, with the tendency to eliminate the derivatization for the application of internal standards marked isotopically and the use of a calibration line built into the matrix solution.

### 2.2. LOD and LOQ

In the procedure for GLY extraction in WR, the LOD was 0.0027 mg kg^−1^, and the LOQ was 0.0092 mg kg^−1^ (the lowest fortification level, which achieved acceptable accuracy). In the procedure for GLY extraction in BR, the LOD was 0.0136 mg kg^−1^, and the LOQ was 0.0456 mg kg^−1^. These values obtained in our study were lower than 0.1 mg kg^−1^, the lower limit of analytical determination, established in EU No. 293/2013 [8]. 

There are several published methods to detect GLY in cereals and several analytical techniques (Table 3). Our values, validated for WR analysis, were similar to those obtained by Santillo et al. (2019) [31] and Nagatomi et al. (2013) [27] in rice and maize and the last one in corn achieved by LC-MS/MS. While our limits found in the BR procedure were similar to those reported by Chamkasem and Harmon (2016) [28] in corn with LC-MS/MS application, they were lower than the other LODs mentioned by Mol and van Dam (2014) [25] in wheat flour by Flow injection–MS/MS, by Herrera López et al. (2019) [32] in oat by LC-MS/MS, and by Gotti et al. (2019) [23] in wheat by CE-UV.

## 3. Material and Methods

### 3.1. Chemical and Reagents

Glyphosate, aminomethylphosphonic acid, and malonic acid were purchased from Merck (Darmstadt, Germany). Ammonia solution (30%) and methanol were purchased from Carlo Erba (Milan, Italy). Standard solutions of phosphorous (10,000 mg L^−1^) were obtained from Inorganic Ventures (Lakewood, NJ, USA).

HPLC-grade water was obtained from an ultrapure water purification system (18.2 MΩ cm, ELGA PURELAB flex, Veolia Water Solutions and Technologies, Ontario, Canada). 

### 3.2. Chromatographic and Mass Spectrometry Conditions

Separation was performed on a Hamilton PRP-X100 column (250 × 2.1 mm, 5 μm particle size), which was installed in an Agilent 1260 Bio-inert LC system, consisting of a G5654A Bio-inert pump and G5668A Bio Multisampler (Agilent Technologies, Santa Clara, CA, USA). The column was maintained at a controlled temperature for all analysis. The outlet of the LC column was connected directly to the ICP-MS nebulizer. The operating conditions for the LC are shown in Table 4. For the determination of GLY as P with ICP-MS/MS (8900, Agilent Technologies), O_2_ was used as a reaction gas in the octopole reaction cell. The Q1 was set to transmit m/z 31, and the Q2 was selected to monitor m/z 47, so that the product ion ^31^P^16^O^+^ could be detected free of any interferences. 

### 3.3. Sample Preparations

Rice grain samples, divided between white rice samples (WR) and brown rice (BR), were ground with a blender. Then, 2.0 g of pulverized rice grains were extracted using 20 mL of ultrapure water with 30% of methanol in centrifuge tubes. The centrifuge tubes with samples were subjected to mechanical agitation for 60 min and ultrasounds for 15 min. Then, the samples were centrifuged at 2044× *g* for 10 min. The supernatant was recovered, filtered through an 0.2 μm nylon filter, and inserted into HPLC vials. 

### 3.4. Method Validation

The quantification of GLY in different types of rice was validated in terms of the linearity, trueness, precision, LOD, LOQ, linearity, and the matrix effect. 

### 3.5. Accuracy and Precision

The trueness of the method was calculated as the percentage recovery of the GLY from the fortified WR spiked at 0.01, 0.03, and 0.05 mg kg^−1^, expressed as GLY, and from the fortified BR samples spiked at 0.05, 0.14, 0.27, mg kg^−1^, with the addition of appropriate volumes of the working standard solution.

Three replicates were prepared and analyzed for each level, according to the methods described. 

The precision in the case of repeatability [relative standard deviation repeatability (RSDr)] was determined by analyzing five replicated samples at each fortification level, for both white and brown rice) on the same day.

### 3.6. LOD and LOQ

In order to determine the LOD for each analyte, 10 independent sample blanks fortified at 0.01 for WR and 0.05 mg kg^−1^ for BR were injected, and it was expressed as the analyte concentration corresponding to three times the standard deviation. 

The LOQ was determined as the lowest fortification level for glyphosate, for which there was acceptable accuracy in terms of trueness (mean recoveries in the range of 70–120%) and precision (RSDr < 20%).

### 3.7. Calibration Curve and Linearity

Calibration curves were made by plotting the mean peak area of 31P → 47 versus concentration, expressed as the molecule concentration. The linearity was evaluated by calculating the coefficient of determination (R^2^), intercept, and slope of the regression line at 0.5, 1, 5, and 10 μg GLY L^−1^ in the WR extraction solution and 20, 50, and 100 μg GLY L^−1^ in the BR extraction solution.

Each calibration solution was prepared by dilution of the GLY stock solution. 

### 3.8. Matrix Effects

The response of the target analyte may be enhanced or suppressed compared to the solvent-based standards. To assess the matrix effect, we developed calibration curves in extraction solutions from two types of rice free of GLY. These effects were estimated by comparing an analyte’s ICP-MS/MS response at any given concentration of the spiked post-extraction sample to a spiked concentration in the solvent (standard solutions). We calculated the matrix effect, as in Chamkasem et al. (2016) [28], applying this expression: 

ME = (slope of the calibration curve of the analyte in the sample matrix/slope of the calibration curve of the analyte in the solvent matrix) × 100. 

A value of 100% means that no matrix effect was present. If the value was less than 100%, it means that there was matrix suppression, but if the value was more than 100%, it means that there was matrix enhancement.

## 4. Conclusions

There are several methods for GLY analysis in different types of cereals; most of these procedures are laborious and include various purification steps. Derivatization is not always applied, and the development of new methods to reduce the pretreatment steps in the analysis of polar molecules is of interest to the scientific community. The present study’s method is faster than the other options, as it does not require pre-cleanup or derivatization, while it is good enough to detect and quantify GLY. These results were possible by using a metal-free liquid chromatography system, with the absence of iron and steel in the solvent delivery lines, and using an anionic column designated to separate the different chemical species containing phosphorus with 2.1 mm of diameter. ICP-MS/MS is a robust detector with a reaction cell that chemically resolves ^31^P^+^ quantification from polyatomic ions with oxygen gas. The matrix effect is well known as one of the critical problems for pesticide residue analysis, and the quantification procedures, performed using the calibration curves constructed in matrices as a reference and taking into account the response of the corresponding matrix, are a reliable way to avoid artefact results.

## Figures and Tables

**Figure 1 molecules-27-08049-f001:**
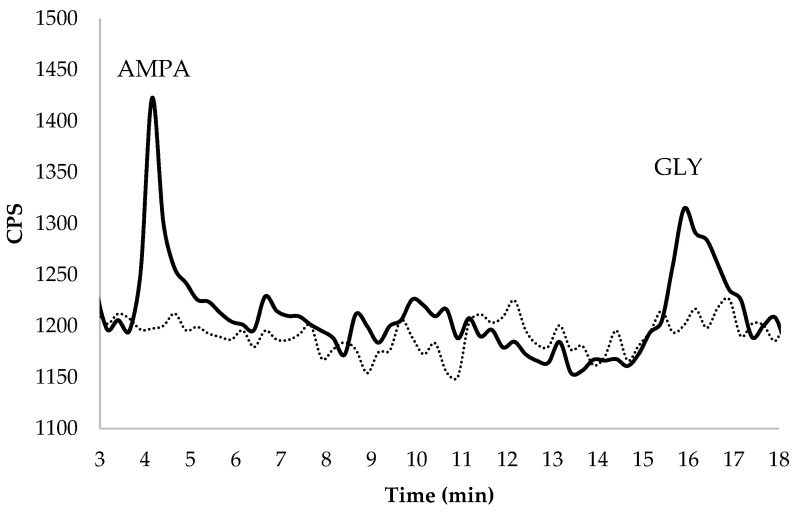
Chromatographic separation in spiked groundwater at 2.7 μg GLY L^−1^ and 1.8 μg AMPA L^−1^ of the target analytes (RT_AMPA_ = 4.15 min; RT_GLY_ = 16.5 min). The dashed line represents the chromatogram of unspiked water.

**Figure 2 molecules-27-08049-f002:**
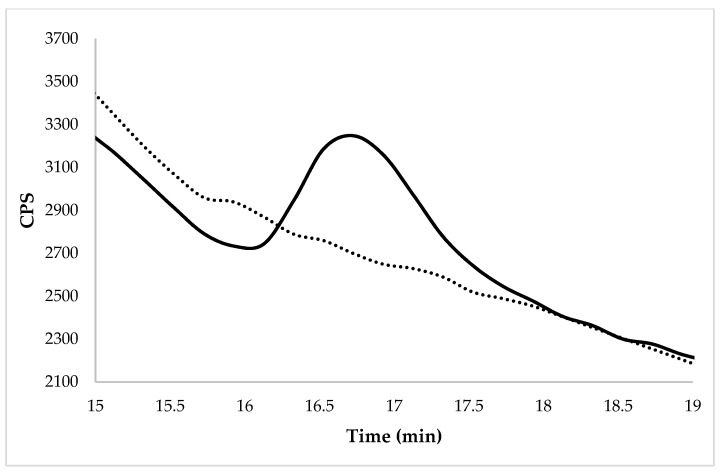
Chromatogram of WR (Caravaggio variety) at the spiked level of 0.01 mg kg^−1^ by HPLC-ICP-MS/MS. The dotted line represents the same un-enriched sample.

**Figure 3 molecules-27-08049-f003:**
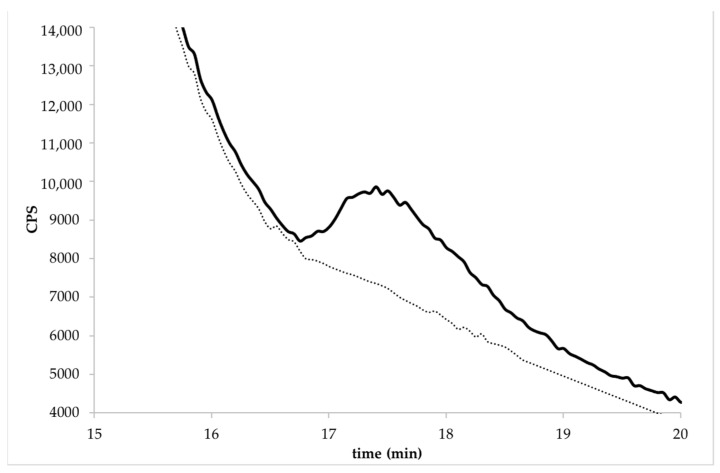
Chromatogram of BR (Barone variety) at the spiked level of 0.1 mg kg^−1^ by HPLC-ICP-MS/MS. The dotted line represents the same un-enriched sample.

**Figure 4 molecules-27-08049-f004:**
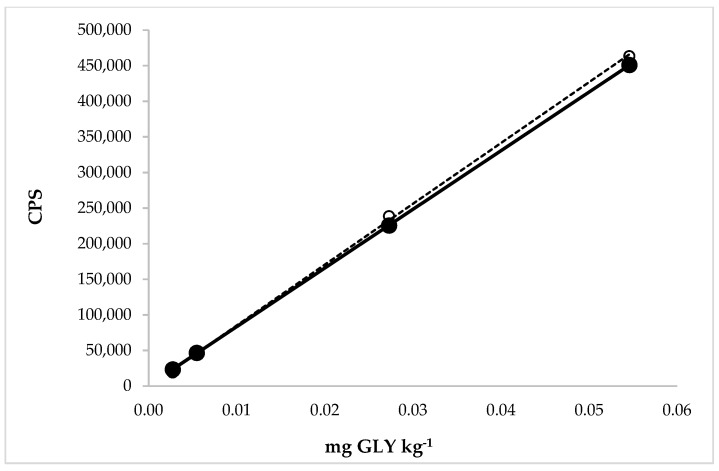
Comparison between the calibration curves of GLY in the extraction solution (dotted line) with linear equation y = 8.543 × 10^6^x − 244.4 and regression fit 0.999 and in the extraction solution with the matrix (white rice, solid line) with linear equation y = 8.240 × 10^6^x + 1100 and regression fit 1.000.

**Figure 5 molecules-27-08049-f005:**
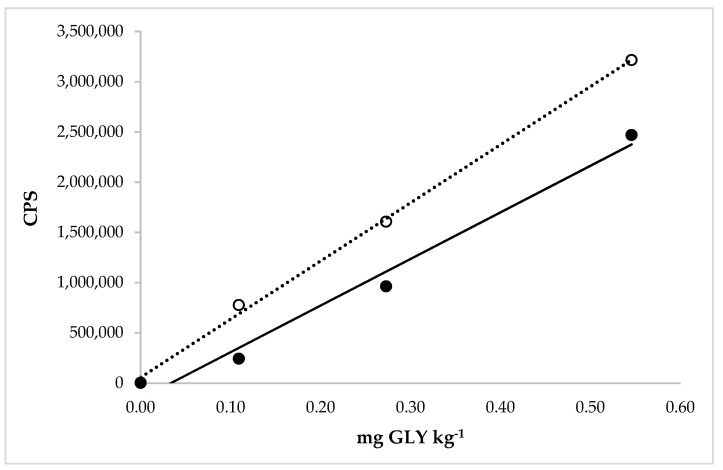
Comparison between the calibration curves of GLY in the extraction solution (dotted line) with linear equation y = 5.789 × 10^6^x + 57,500 and regression fit 0.998 and in the extraction solution with the matrix (brown rice, solid line) with linear equation y = 4.636 × 10^6^x − 155,600 and regression fit 0.982.

**Table 1 molecules-27-08049-t001:** Average recoveries (%), standard deviation (SD), and coefficient of variation (CV%) at different spiked levels of the extracted solution from white and brown rice. † Three replicates for each level. ‡ Five replicated samples at each fortification level.

Matrix	Spiked Levels (mg kg^−1^)	Recovery (%) ^†^	Standard Deviation (SD)	Coefficient of Variation (CV%) ^‡^
White rice	0.01	76	8	11
	0.03	90	6	6
	0.05	105	3	3
Brown rice	0.05	94	8.3	8.8
	0.14	99	1.3	1.4
	0.27	97	2.6	2.7

**Table 2 molecules-27-08049-t002:** Literature on the glyphosate determination in cereals, with details of the matrix effect.

Articles	Matrix	Instrument	Derivatisation	Isotopic Internal Standard	Clean up	ME
Botero-Coy et al., (2013) [26]	rice	LC-MS/MS	/	ILIS-isotope labeled GLY	/	70–80%
Botero-Coy et al., (2013) [26]	maize	LC-MS/MS	/	ILIS-isotope labeled GLY	OASIS HLB	75%
Nagatomi et al., (2013) [27]	corn	LC-MS/MS	/	/	OASIS MCX + INERT SEK	only mention
Mol and van Dam 2014 [25]	wheat flour	FI-MS/MS ^c^	/	yes	/	20 < ME < 40
Chamkasem and Harmon, (2016) [28]	corn	LC-MS/MS	/	yes	OASIS HLB	101%
Liao et al. (2018) [29]	rice, wheat, maize	LC-MS/MS	FMOC-Cl ^a^	yes	SPE-C18 cartridge (60 mg)	/
Zoller et al., 2018 [30]	wheat, white flour	LC-MS/MS	/	yes	SPE cartridge	no indication of disturbing matrix effects
Gotti et al., 2019 [23]	wheat	CE-UV	FMOC-Cl ^a^	Taurine ^b^	SPE C18 cartridge SAX cartridge	no significant differences were found
Herrera López et al. (2019) [32]	oat	LC-MS/MS	/	ILIS-isotope labeled GLY	/	34%
Santilio et al., (2019) [31]	rice	LC-MS/MS	/	yes	/	77%
Santilio et al., (2019) [31]	maize	LC-MS/MS	/	yes	/	104%
Ciasca et al., 2020 [24]	wheat	FI-MS/MS ^c^	/	yes	OASIS HLB	no effect
This study	white rice	HPLC-ICP-MS/MS	/	^89^Y	/	96%
This study	brown rice	HPLC-ICP-MS/MS	/	^89^Y	/	80%

^a^ 9-fluorenylmethylchloroformate. ^b^ internal standard for CE-UV. ^c^ flow injection tandem mass spectrometry.

**Table 3 molecules-27-08049-t003:** Literature on the glyphosate determination in cereals, with details on the LOD and LOQ.

Articles	Matrix	Instrument	Fortification Level of Matrix	LOD	LOQ
Botero-Coy et al., (2013) [26]	rice	LC-MS/MS	0.1 mg GLY kg^−1^	0.008 mg GLY kg^−1^	0.03 mg GLY kg^−1^
Botero-Coy et al., (2013) [26]	maize	LC-MS/MS	0.1 mg GLY kg^−1^	0.007 mg GLY kg^−1^	0.02 mg GLY kg^−1^
Nagatomi et al., (2013) [27]	corn–malt	LC-MS/MS	0.010 mg GLY kg^−1^		0.010 mg GLY kg^−1^
Mol and van Dam (2014) [25]	wheat flour	FI-MS/MS *	0.2 mg GLY kg^−1^		0.1 mg GLY kg^−1^
Chamkasem and Harmon, (2016) [28]	corn	LC-MS/MS	0.1 mg GLY kg^−1^	0.015 mg GLY kg^−1^	0.045 mg GLY kg^−1^
Liao et al. (2018) [29]	rice, wheat, maize	LC-MS/MS	0.005 mg GLY kg^−1^	0.0017 mg GLY kg^−1^	0.005 mg GLY kg^−1^
Zoller et al., 2018 [30]	wheat, white flour	LC-MS/MS	0.001 mg GLY kg^−1^	0.0003 mg GLY kg^−1^	0.001 mg GLY kg^−1^
Gotti et al., 2019 [23]	wheat	CE-UV			0.1 mg GLY kg^−1^
Herrera López et al. (2019) [32]	oat	LC-MS/MS	0.1 mg GLY kg^−1^		0.1 mg GLY kg^−1^
Santilio et al., (2019) [31]	rice	LC-MS/MS	0.01 mg GLY kg^−1^	0.002 mg GLY kg^−1^	0.01 mg GLY kg^−1^
Santilio et al., (2019) [31]	maize	LC-MS/MS	0.01 mg GLY kg^−1^	0.004 mg GLY kg^−1^	0.01 mg GLY kg^−1^
Ciasca et al., 2020 [24]	wheat	FI-MS/MS *			2.0 mg GLY kg^−1^
This study	white rice	HPLC-ICP-MS/MS		0.0027 mg GLY kg^−1^	0.0092 mg GLY kg^−1^
This study	brown rice	HPLC-ICP-MS/MS		0.0136 mg GLY kg^−1^	0.0456 mg GLY kg^−1^

* flow injection tandem mass spectrometry.

**Table 4 molecules-27-08049-t004:** The operating conditions of the anion exchange chromatography combined with inductively coupled mass spectrometry triple quadrupole (ICP-MS/MS).

HPLC Parameters	
HPLC	Agilent 1260 Bio-inert LC system
Column:	Hamilton, PRP-X100, 250 × 2.1 mm, 5 μm particle size
Column temperature:	50 °C
Mobile phase:	2 mM malonic acid (C_3_H_4_O_4_) at pH 5.3
Injection volume:	60 μL
Flow rate:	0.6 mL min^−1^
Acquisition time:	25 min
ICP-QQQ-MS	
Scan mode	MS/MS
RF applied power (W)	1550
Sampling depth (mm)	8.0
Lens type	s-lens
Octopole bias (V)	−5.0
Octopole RF (V)	180
KED (V)	0
Cell reaction gas	O_2_
Cell gas flow rate (%)	25
Cell entrance (V)	−50
Cell exit (V)	−70
Deflect (V)	6.0
Plate bias (V)	−60
Monitored mass (Q1)	31–60
Monitored mass (Q2)	47

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
