# Peer review of "Determination of Glyphosate in White and Brown Rice with HPLC-ICP-MS/MS"

_molecules, 2022, doi:10.3390/molecules27228049_

Round 1
Reviewer 1 Report
The presented manuscript describes the quantitative analysis of glyphosate in rice by the HPLC-ICP-MS/MS technique. This study was carried out correctly, following the art. I also have no objections to the way the results are presented. Only, looking at the significant broadening of the signal with the retention time of 16.5 minutes in figure 1, I have doubts about the correctness of the detection limit. In general, the article reads well and fluently. However, my complaint is about the lack of novelty. The determination of glyphosate using this technique and a similar approach has already been described. Therefore, I do not recommend this manuscript for publication in Molecules.
Reviewer 3 Report
Fontanella and Beone reported a LC-ICP-MS?MS method for analyzing Gly in white and brown rice. The article is well written. I have the following comments that I would like the authors to address:
1. Please use full form while the abbreviation is used for the first time i.e., in the abstract.
2. It looks like the LOD was so different between WR and BR due to the high matrix effect in the latter. It would be advised to assess different established approaches (e.g. solid phase extraction, LLE, protein precipitation, or any others which the authors listed in table 3) to minimize the matrix effect which will demonstrate the application of the developed method better. I understand that the authors tried to develop a method with a reduced number of steps, however high matrix effects in BR might warrant using a matrix-matched calibration which ultimately increases a step anyhow.
3. The authors could try different LLEs. Using chloroform/hexane can help with minimizing the matrix effect, especially the one seen in BR.
4. Introduction has so many paragraphs, please try to reduce having paragraphs with one sentence.
5. Please replace dangerousness with a better word that fits.
6. Please state that P stands for phosphorus on page 2 line 90.
7. The LOQ for GLY in the reported study on page 3 is 0.29 ug/L which is different than what has been mentioned in table 4, can the authors clarify that a bit?
8. It appears the dynamic range of linearity was tested on 4 concentration points. I would recommend increasing the number of concentrations tested. Also, how the mg/Kg translates to ug/L. It would be better to use the same unit which will be helpful for the readers to follow and utilize in their studies.
Round 2
Reviewer 1 Report
Dear Authors,
Based on the answer obtained, I am inclined to recommend the article for publication. I have no additional objections.